# Effects of *Irvingia gabonensis* Extract on Metabolism, Antioxidants, Adipocytokines, Telomere Length, and Aerobic Capacity in Overweight/Obese Individuals

**DOI:** 10.3390/nu14214646

**Published:** 2022-11-03

**Authors:** Rujira Nonsa-ard, Ploypailin Aneknan, Terdthai Tong-un, Sittisak Honsawek, Naruemon Leelayuwat

**Affiliations:** 1Biomedical Sciences Program, Graduate School, Khon Kaen University, Khon Kaen 40002, Thailand; 2Exercise and Sport Sciences Development and Research Group, Khon Kaen University, Khon Kaen 40002, Thailand; 3Department of Physiology, Faculty of Medicine, Khon Kaen University, Khon Kaen 40002, Thailand; 4Center of Excellence in Osteoarthritis and Musculoskeleton, Faculty of Medicine, Chulalongkorn University, Bangkok 10330, Thailand; 5Graduate School, Khon Kaen University, Khon Kaen 40002, Thailand

**Keywords:** *Irvingia gabonensis*, ascorbic acid, malondialdehyde, adiponectin, body mass index, substrate utilization, maximum oxygen consumption, biomarker of aging

## Abstract

We investigated the effects of *Irvingia gabonensis* (IG) kernel extract on the metabolism, adiposity indices, redox status, inflammation, adipocytokines, blood leukocyte relative telomere length (RTL), and aerobic capacity of overweight/obese individuals. All participants used the first 12-week phase to monitor body weight. They were then randomly divided into two groups: (1) 300 mg IG or (2) placebo (PLA). Both groups took one tablet per day for 12 weeks. The variables were measured before supplementation and after 3, 6, and 12 weeks of supplementation. RTL and aerobic capacity were measured before and after 12 weeks. Compared with the PLA, the IG increased plasma vitamin C after supplementation at 6 (*p* < 0.01) and 12 weeks (*p* < 0.05) and serum adiponectin after 3 weeks (*p* < 0.05). Compared with before supplementation, plasma malondialdehyde in the IG and serum leptin in the PLA were decreased after 12-week supplementation, without any differences between the groups. There were no differences between groups with respect to metabolism, inflammation, RTL, and aerobic capacity after the supplementation. We suggest that 12-week daily IG supplementation improved plasma vitamin C and adiponectin. The findings show the possible mechanism contributing to the effect of IG supplementation on a reduction in obesity-related complications.

## 1. Introduction

Overweight and obesity are commonly defined by body mass index (BMI). In the Asia-Pacific region, overweight and obesity are defined as, respectively, a BMI between 23 and 24.9 kg/m^2^ and BMI greater than 25 kg/m^2^ [1]. Overweight and obesity have become worldwide health problems [2]. They can lead to a broad spectrum of diseases, especially cardiovascular diseases, type 2 diabetes mellitus, and osteoarthritis [3].

Adipose tissue dysfunction, commonly found in people who are overweight or obese, leads to an increase in proinflammatory cytokines such as C-reactive protein (CRP), which induces low-grade inflammation [4]. Obesity-induced inflammation promotes the overproduction of reactive oxygen species (ROS) [5]. ROS can cause damage to a variety of biological structures, including lipids, proteins, and DNA, particularly in the telomere region [6]. Moreover, these alterations of adipose tissue cause abnormal secretion of adipokines, especially adiponectin and leptin, which play an important role in energy regulation [7]. Furthermore, obesity impairs fat oxidation during rest [8], at submaximum and peak intensity, and during recovery [9]. Decreased fat oxidation and exercise capacity may result from the appearance of adipokine abnormalities [10].

Apart from reducing energy intake and increasing physical activity, dietary supplements may help people who are overweight or obese to achieve weight loss. To the best of our knowledge, supplements of plant-based extracts such as *Garcinia cambogia* [11], Celastrol [12], grape seed proanthocyanidin [13], and *Vernonia mespilifolia* Less. [14] have been used to reduce weight and the pathological changes associated with obesity. Another extract that has become a popular herbal dietary supplement for weight loss is African mango [15], whose scientific name is *Irvingia gabonensis* (IG) (from the *Irvingiaceae* family).

Although some meta-analysis studies have reported that IG extract has some potential benefit for weight loss [16,17], its effects on obesity-related pathologies such as adiposity indices, inflammation, and circulating metabolic markers in overweight or obese subjects are inconclusive [18,19,20]. Ngondi et al. performed two studies of IG seed extract in 2005 [19] and 2009 [20]. The former study investigated the intake of 1.05 g of IG extract three times a day, half an hour before meals, for four weeks in obese subjects. Significant effects were observed in respect of IG only in the reduction of body weight, without any significant effects on waist circumference, blood glucose, or lipid profile compared to PLA [19]. However, the latter study, in which 150 mg IG seed extract was taken twice daily, 30–60 min before lunch and dinner, for 10 weeks, showed improvements in body weight, body fat, and waist circumference as well as plasma total cholesterol, low-density lipoprotein cholesterol, blood glucose, and high sensitivity C-reactive protein (hsCRP) in overweight and/or obese individuals [20]. However, neither study reported appropriate research methodologies. Although the studies involved randomized controlled trials (RCTs), they did not report appropriate randomization or allocation concealment. Neither of the studies performed sample size calculation and an intention-to-treat analysis. Furthermore, subjects in the study of Ngondi et al. (2005) had different characteristics at baseline.

In addition, no studies have been undertaken concerning the effects of IG seed extract on the redox status, fat oxidation rate, telomere length (TL), and aerobic capacity of overweight/obese individuals. A recent review reported that IG fruits contain higher ascorbic acid concentrations than some vitamin-C-rich fruits [21]. These data are comparable with in vitro studies showing that IG seed extract has beneficial effects on antioxidant activity, free radical scavenging, and lipid peroxidation [22,23]. Regarding adipocytokines, IG supplementation has been shown in an in vitro study to downregulate the leptin gene and upregulate the adiponectin gene [24]. Ngondi et al. (2009) confirmed improvements in adipocytokines in both overweight and/or obese individuals [20]. However, their study may not be sufficiently valid. Therefore, high-quality RCT studies that investigate the effects of IG on adipocytokines and other variables such as redox status, inflammation, fat oxidation rate, TL, and aerobic capacity in overweight or obese subjects are necessary.

This study was primarily designed to examine the effects of IG kernel extract on the fat oxidation rate in overweight or obese individuals. Adiposity indices, circulating metabolic markers, redox status, adipocytokines, inflammation, TL, and aerobic capacity were also measured. This information may help in suggesting alternative strategies by which to prevent and treat non-communicable metabolic diseases, especially in overweight/obese individuals. It was hypothesized that IG kernel extract for 12 weeks would improve fat oxidation rate, adiposity indices, circulating metabolic markers, redox status, adipocytokines, inflammation, TL, and aerobic capacity in overweight or obese individuals.

## 2. Materials and Methods

### 2.1. Participants

The sixty-six participants in the study were women with overweight/obesity who had a BMI between 23 and 29.9 kg/m^2^ (based on the Asia-Pacific criteria) [1], aged between 30 and 59 years. The study was conducted in Khon Kaen province, Thailand, and nearby, from August 2020 to September 2021. Posters and direct invitations were used for recruitment on campus and in the community. Before providing written informed consent, all participants were briefed on the purpose of the study, both orally and in writing.

Participants were recruited if they had stable weight, had not attempted a weight reduction program, had not taken any supplements for at least three months before recruitment, had a normal lipid profile or abnormal lipid profile without medicine, had systolic blood pressure (SBP) ≤ 140 mmHg and diastolic blood pressure (DBP) ≤ 90 mmHg, did not smoke, and did not drink alcohol. Participants were excluded if they were pregnant; had fasting blood glucose (FBG) ≥ 126 mg/dL, serum glutamate pyruvate transaminase (SGPT) > 35 IU/L, creatinine (Cr) > 1 mg/dL; or had been diagnosed with neuromuscular diseases, orthopaedic problems, autonomic neuropathy, chronic infections, endocrine diseases, or drug or natural ingredient allergies. Lipid profile, Cr, and SGPT were measured using a Reflotron Plus (Roche, Boehringer Mannheim, Germany). BP was measured using an automatic sphygmomanometer bedside monitor (SVM-7600 Nihon Kohden, Malaysia). FBG was measured using the glucose oxidase method using a Glucose and l-Lactate Analyzer (YSI Incorporated, Yellow Springs, OH, USA). The study was conducted in accordance with the Declaration of Helsinki and approved by the Ethics Committee of the Center for Ethics in Human Research, Khon Kaen University (HE621385, date approved: 26 February 2020). This research project was approved for registration at the Thai Clinical Trial Registry (TCTR) (TCTR20200803005, date approved: 2 August 2020).

### 2.2. Research Design, Randomised Allocation, and Blinding

The 12-week randomized, double-blind, placebo-controlled, pre- and post-test study was performed at Khon Kaen University, Thailand. The randomized allocation sequence 1:1 was generated using computer-generated random numbers and kept in sequentially numbered, opaque, sealed envelopes. Participants received supplement tablets from Blackmores Limited labelled in a randomized sequence throughout the study. To maintain and guarantee blinding, both products had identical shape, size, color, smell, and packaging. The participating investigators, medical covering staff, and the participants were blinded to group allocation.

#### 2.2.1. Power Calculation

The sample size of this study was calculated based on fat oxidation rate in a study by Robert et al., 2015 [25]. They reported the role of decaffeinated green tea extract in increasing the fat oxidation rate in 14 active males. In the current study, we applied the mean (μ1 = 0.241 g/min, μ2 = 0.301 g/min) and standard deviation (SD) (SD1 = 0.094, SD2 = 0.065) of the fat oxidation rate using the formula of Rosner (2015) [26]. A total of 72 participants (including 20% dropout) scored 0.80 for power of test and 0.70 for effect size.

#### 2.2.2. IG and PLA Preparation

Both IG and PLA tablets were provided by Blackmores Limited (Warriewood, NSW, Australia). IG tablets contained 300 mg of IG kernel extract. PLA tablets contained mainly microcrystalline cellulose (615 mg) and maltodextrin (115 mg). Both types of tablets were kept in bottles throughout the 12 weeks. The products were provided to the participants at the start of the supplementation phase (visit 2).

The kernel extract of IG (trade name IGOB131) in this study has previously been evaluated for safety. This is confirmed by the IGOB131 specification sheet (NSF certification printed on 21 September 2018). In addition, the manufacturer’s products are of high-quality, as indicated by the Good Manufacturing Practice (GMP) registration in respect of dietary supplements (NSF certification). Both documents are shown in the Appendix A. Furthermore, in vitro and in vivo chromosomal aberration tests, and an in vivo micronucleus assay, did not show any genotoxicity of IGOB131 [27]. The results of a sub-chronic toxicity study also suggested that the no-observed-adverse-effect level (NOAEL) for IGOB131 is ≥2500 mg/kg body weight (BW)/day, which was the highest dose tested [27].

#### 2.2.3. Supplementation Intake and Adverse Event Monitoring

To monitor supplementation intake and adverse events, we gave the participants a form on which to record the number of tablets taken and any adverse events. We also telephoned them once a week for this information. If they forgot to take the tablet, they were told to skip that day. At the last visit (visit 5), they returned the bottle containing unused tablets and the form they used to record their intake and adverse events. Their liver and kidney functions were monitored via plasma Cr and serum SGPT concentrations.

### 2.3. Study Protocol

In this study, all the participants made 5 visits over two 12-week phases: the BW monitoring phase and the supplementation phase (Figure 1). At visit 1, participants were screened, and if they passed, they participated in the BW monitoring phase. At the end of the monitoring phase (visit 2), if they were able to maintain a change of BW not greater than four kilograms, they were randomly divided into two groups, receiving one 300 mg tablet daily of either (1) IG or (2) PLA. Variables were measured at 4 points: before supplementation (visit 2) and after supplementation for 3 weeks (visit 3), 6 weeks (visit 4), and 12 weeks (visit 5). The blood leukocyte relative telomere length (RTL) was represented by the T/S ratio, which is the ratio of telomeric repeats to a single copy gene, the 36b4 gene. RTL and aerobic capacity were measured only at visit 2 and visit 5 (Figure 1). All the participants performed the experiment during the follicular phase of the menstrual cycle.

#### Blood Collection

At the screening visit (visit 1), 5 mL whole blood was drawn from the antecubital vein into 3 tubes after a 12-h overnight fast. A total of 1 mL whole blood was kept in a sodium fluoride tube for FBG measurement. A total of 3 mL blood was drawn into an ethylenediaminetetraacetic acid (EDTA) tube for lipid profile and Cr measurement. The last 1 mL whole blood was kept in a serum separator gel tube for SGPT measurement. At visits 2–5, in addition to the measurements taken at visit 1, a 10 mL whole blood sample in an EDTA tube was used for measuring the lipid profile, Cr, and MDA concentration. Furthermore, 6 mL whole blood in a serum separator gel tube was used for measuring serum hsCRP, leptin, and adiponectin. A total of 3 mL blood was drawn into a lithium heparin tube, which was wrapped in aluminum foil to block out light, to measure plasma vitamin C. A total of 500 μL of plasma in this tube and HClO_4_ (1 mol/L) were mixed for protein precipitation. Moreover, the buffy coat collected was used for measuring RTL.

The blood samples were centrifuged at 3000 rpm, at 4°C, for 10 min (TOMY-CAX-370, Tokyo, Japan) and aliquots of plasma, serum, and buffy coat samples were stored immediately at −80°C until measurement took place.

### 2.4. Outcome Measurements

#### 2.4.1. Adiposity Outcomes

BW and height (Ht) were measured using a stadiometer (DETECTO, St. Webb City, MO, USA). The participants stood with bare feet together, the upper part of the back not touching the scale and wearing minimal clothing. BW and Ht were used to calculate BMI. W and hip (H) circumferences were measured using non-elastic tape at the midpoint between the lower rib margin and the iliac crest during full expiration, and around the widest part of the hip in the standing position, respectively, to the nearest 0.1 mm. The coefficient of variation (CV) of the anthropometry measurement was 0.3%. Fat mass (FM) was measured using dual-energy X-ray absorptiometry (DXA) (Lunar Prodigy whole-body scanner, GE Healthcare, Chicago, IL, USA), with participants in a supine position, at Srinagarind Hospital.

#### 2.4.2. Dietary Intake and Energy Expenditure Record

A week before every visit, the participants were required to record their dietary intake and physical activity for three days per week (two during the week and one on the weekend). The records were used to monitor their normal dietary habits and daily physical activity and to evaluate energy intake and expenditure. Energy intake was analyzed using the INMUCAL program (INMUCAL software, Mahidol University, Salaya campus, Nakorn Pathom, Thailand) [28].

#### 2.4.3. Metabolic Markers

##### Fat Oxidation Measurement

To measure the amount of fat oxidized, indirect calorimetry (Oxycon CareFusion 234 GmbH, Höchberg, Germany) was used through the analysis of carbon dioxide and oxygen concentrations in expired gas. Both expired gases were used to calculate the fat oxidation rate following the Peronnet and Massicotte equation with protein ignoring [29]. Gas collection was performed while the subject was resting for 20 min in the supine position, awake, and not using a smartphone.
Fat oxidation rate = 1.695 *V*O_2_ − 1.701 *V*CO_2_

##### Blood Chemistry

FBG was measured using a Glucose Analyzer (YSI Incorporated, Yellow Springs, OH, USA). Plasma lipid profile, Cr, and serum SGPT were measured using a Refloton Plus (Roche, Boehringer Mannheim, Germany). The CV of blood chemistry measurement using the Refloton Plus was 0.1%. At every run, the machine was calibrated by a technician following factory instructions. All these parameters were measured within a day after blood drawing.

##### Serum Leptin and Adiponectin Measurements

Serum leptin was tested using a leptin (human) EIA kit (Bertin Technologies, Montigny-le-Bretonneux, France) based on the sandwich technique. Leptin was bound using a polyclonal antibody coated on the plate and detected using a second polyclonal antibody tagged with a leptin receptor antibody (HRP) that is specific for leptin. The concentrations were determined by measuring the enzymatic activity of the immobilized tracer using 3,3′,5,5′-tetramethylbenzidine (TMB) to form a yellow compound after the reaction had been stopped. The intensity of color was determined using spectrophotometry at 450 nm. The average inter- and intra-assay CVs were 13% and 5% in duplicate of serum leptin measurement. Serum adiponectin was measured using an adiponectin (human) EIA kit (Bertin Technologies, Montigny-le-Bretonneux, France). The concentration of serum adiponectin was measured using a competitive EIA technique. Hydrogen peroxide/TMB substrates were used to determine the competition between free and coated adiponectin, in the presence of a known quantity of HRP-labelled antibody (tracer). The tracer acted on the substrate to form a yellow compound that absorbs at 450 nm. The average inter- and intra-assay CVs were 12% and 8% in duplicate serum adiponectin measurements.

#### 2.4.4. Redox Status Measurement

Lipid peroxidation was detected through the reaction of MDA with thiobarbituric acid (TBA) at low pH and high temperature to form a colored complex, i.e., the MDA-TBA complex. The absorbance was measured at 532 nm using visible absorption spectrophotometry, as described in Draper et al. [30]. The average inter- and intra-assay CVs were 8% and 7% in duplicate MDA measurement. 

Plasma vitamin C concentration was measured using Zhang’s method [31]. Potassium ferricyanide (K_3_[Fe(CN)_6_]) was used as the spectroscopic probe reagent of ascorbate. The ascorbate was oxidized to dehydroascorbic acid, which subsequently reacted with potassium ferricyanide to form soluble Prussian blue. The solution’s absorbance was measured at 735 nm. The average inter- and intra-assay CVs were 6% and 9% in duplicate plasma vitamin C measurement.

#### 2.4.5. Inflammation Measurement

hsCRP was measured at Srinagarind Hospital using the Cobas^®^ 6000 Analyzer Series (Hitachi, Mannheim, Germany). At every run, the machine was calibrated by a technician following factory instructions. hsCRP was measured within a day after blood drawing.

#### 2.4.6. RTL Measurement

DNA was extracted using a FlexiGene DNA kit (FlexiGene DNA kit; Qaigen, Hilden, Germany). RTL was then measured using quantitative real-time polymerase chain reaction (qPCR) adapted from the method originally described by Cawthon [32] and previously reported [33]. In brief, RTL was shown as a proportion between telomeric repeat (T) and a single copy gene (S), i.e., the 36b4 gene T/S ratio. Both PCRs were activated in 10 µL as the final volume with a PowerUp™ SYBR™ Green Master Mix (2x) (Applied Biosystems, Foster City, CA, USA) by a QuantStudio™ 6 Flex Real-Time PCR System (Applied Biosystems, Foster City, CA, USA). The intra- and inter-assay coefficient of variation for RTL measurement was 1.0% for this study.

#### 2.4.7. Aerobic Capacity Test

The maximum oxygen consumption (*V*O_2max_), indicated by aerobic capacity, was measured using an incremental graded exercise test [34]. This test was performed on a cycle ergometer until the heart rate (HR) reached 85% of the maximum HR, which was calculated using the formula developed by Tanaka et al. (2001) [35].

### 2.5. Statistical Analysis

The data were analyzed using Statistical Package for the Social Sciences (SPSS) for Windows, version 28.0 (IBM Corp., Armonk, NY, USA). The normality was tested using the Shapiro–Wilk test. Mean differences of all baseline variables were tested using the Man–Whitney U-test and an independent *t*-test based on data normality. A generalized estimating equation (GEE) was used for estimating the effects of IG extract and PLA over the duration of the study. A Bonferroni post hoc test was then used to calculate the significant difference, and results are presented as the mean difference and 95% confidence interval (CI). To adjust with pre-test data, analysis of covariance (ANCOVA) was used to analyze the differences between post-supplementation of *V*O_2max_ and RTL. All data are expressed as mean ± standard deviation (SD) or median (interquartile range, IQR) based on a normal distribution. The significance level was less than 0.05.

## 3. Results

From a total of 96 applicants (95 women and 1 man), 72 participants (all women) were recruited. There were 20 individuals who did not meet the inclusion criteria, and 4 individuals did not respond to the first visit invitation. The 72 participants were allocated to the first phase (12-week body weight monitoring phase) (Figure 2). At the end of the monitoring phase, 1 individual declined to participate for medical reasons and 2 individuals declined to participate for personal reasons. Therefore, there were 69 participants who completed the monitoring phase. They were then randomized to the 12-week supplementation phase. Thirty-four and thirty-five participants were allocated to the PLA and IG groups, respectively. During the supplementation phase, one participant in the PLA group discontinued the intervention because of COVID-19. In the IG group, two participants discontinued the intervention: one did not feel comfortable undertaking DXA, and one had health complications unrelated to the study trial. The final number of participants was 33 in each group. We stopped the recruitment and experiment after receiving 72 participants at screening and 66 participants at the completion of the supplementation phase, because we achieved the sample size that we planned (60 participants excluding 20% dropout).

We were informed of side effects including nausea (6 cases), intestinal constipation (1 case), and thirst (1 case). However, the side effects were not severe and resolved quickly. Interestingly, we found beneficial symptoms such as feeling full (2 cases) and better excretion (1 case). Furthermore, liver and kidney functions indicated by plasma Cr and serum SGPT concentrations were normal throughout the supplementation.

The baseline characteristics of participants who completed the supplementation phase (n = 66) before visit 1 and after the BW monitoring phase (visit 2) are shown in Table 1. The ages of participants in this study were between 38 and 51 years. At the end of the monitoring phase, the participants were able to maintain their body weight to within a four-kilogram range, and all parameters of anthropometry, body composition, and blood chemistry parameters were within the inclusion criteria (Table 1). At the end of supplementation phase, in the PLA group, there were 13 overweight individuals (23–24.9 kg/m^2^) and 20 obese individuals (≥25 kg/m^2^). The same pattern was observed in the IG group, which included 11 overweight and 22 obese individuals. In addition, when comparing with the PLA group, the IG group had similar intakes of carbohydrate (CHO), vitamin C, and energy, and similar energy expenditure, throughout the supplementation phase. However, the IG group consumed less daily fat before and after 3-week supplementation and consumed less daily protein throughout the supplementation phase (*p* < 0.05 and *p* < 0.01) (Table 2).

### 3.1. Adiposity Indices

BMI, W, H, BW, and FM were shown to be not significantly different between groups throughout the supplementation phase (Table 3).

### 3.2. Metabolic Markers

#### 3.2.1. Fat Oxidation

The results show no significant differences in the fat oxidation rate within and between groups at any time points (Table 4).

#### 3.2.2. Blood Chemistry

TG was found to be increased after 12 weeks compared to before administration in the IG group (25.75 mg/dL; 95%CI, 3.54, 47.97; *p* < 0.05) without any significant difference between the groups. There were no significant differences within and between groups in other parameters (Table 5).

#### 3.2.3. Serum Leptin and Adiponectin

In the PLA group, serum leptin concentrations were decreased from before supplementation after taking the product at 6 (−4.75 ng/mL; 95%CI, −8.83, −0.67; *p* < 0.05) and 12 (−3.65 ng/mL;95%CI, −6.97, −0.34; *p* < 0.05) weeks. Moreover, when compared with after 3 weeks’ supplementation, serum leptin concentration was significantly decreased 6 weeks after supplementation (−3.98 ng/mL; 95%CI, −7.34, −0.62; *p* < 0.05) (Figure 3). However, there were no significant changes in serum leptin concentration in the IG group. No significant difference in serum leptin concentration between groups was observed. Furthermore, serum adiponectin concentration was found to be significantly higher in the IG compared to the PLA group after 3 weeks of supplementation (2.22 μg/mL; 95%CI, 0.18, 4.25; *p* < 0.05), but no between-group differences at other time points were found. In the IG group, compared with before supplementation, serum adiponectin concentration was significantly decreased only at 6 weeks (−2.65 μg/mL; 95%CI, −4.73, −0.57; *p* < 0.01), with no changes at other time points during supplementation. In the PLA group, there were significant decreases after taking placebo for 3 (−6.30 μg/mL; 95%CI, −9.14, −3.46; *p* < 0.01), 6 (−5.19 μg/mL; 95%CI, −7.80, −2.58; *p* < 0.01), and 12 (−5.78 μg/mL; 95%CI, −8.49, −3.08; *p* < 0.01) weeks compared with before supplementation. Furthermore, when compared with after 3 weeks’ supplementation, the serum leptin concentration was significantly increased 6 weeks after supplementation (1.11 μg/mL; 95%CI, 0.17, 2.05; *p* = 0.05). Figure 3 provides a more detailed illustration of the effects on serum leptin and adiponectin concentrations in the IG and PLA groups.

### 3.3. Redox Status and Inflammation

Plasma MDA concentrations were found to have decreased after taking both products (PLA group at 6 weeks (−1.54 μmol/mL; 95%CI, −2.78, −0.30; *p* < 0.01) and IG group at 12 weeks (−2.84 μmol/mL; 95%CI, −5.61, −0.07; *p* < 0.05)) without any significant differences between the groups. Compared with after 3 weeks’ supplementation, MDA was found to have decreased at 6 weeks in the PLA group (−2.15 μmol/mL; 95%CI, −4.05, −0.24; *p* < 0.05) and at 12 weeks in the IG group (−2.15 μmol/mL; 95%CI, −4.23, −0.06; *p* < 0.05) (Figure 4). In addition, plasma vitamin C concentrations showed dramatically greater increases in the IG group compared with the PLA group, at 6 (23.45 μmol/L; 95%CI, 11.57, 35.33; *p* < 0.01) and 12 (16.95 μmol/L; 95%CI, 2.80, 31.10; *p* < 0.05) weeks after supplementation. For the IG group, compared with before supplementation, we found significant increases after 6 (28.8 μmol/L; 95%CI, 17.0, 40.6; *p* < 0.01) and 12 (31.0 μmol/L; 95%CI, 18.2, 43.8; *p* < 0.01) weeks of supplementation. We found that the IG group exhibited increases in plasma vitamin C concentrations after 6-week supplementation (25.6 μmol/L; 95%CI, 13.5, 37.6; *p* < 0.01) and 12-week supplementation (27.8 μmol/L; 95%CI, 14.3, 41.2; *p* < 0.01) compared with concentrations after 3-week supplementation. We also found that the PLA group exhibited increases in plasma vitamin C concentrations after 12-week supplementation compared with concentrations before (13.4 μmol/L; 95%CI, 4.92, 21.8; *p* < 0.01), at 3 weeks (10.5 μmol/L; 95%CI, 2.53, 18.5; *p* < 0.01), and at 6 weeks (8.69 μmol/L; 95%CI, 0.65, 16.7; *p* < 0.05) (Figure 4). Moreover, there were no significant differences in hsCRP within and between groups throughout the study (Table 5).

### 3.4. Blood Leukocyte Relative Telomere Length (RTL)

RTL was found not to be significantly different within and between both groups. Before supplementation, the RTL of the PLA group was 0.81 ± 0.27, and at 12 weeks after taking placebo, it was 0.84 ± 0.31. In the IG group, the RTL before supplementation was 0.85 ± 0.38, and at 12 weeks after taking IG, it was 0.81 ± 0.29 (Figure 5).

### 3.5. Aerobic Capacity

There were no significant differences in aerobic capacity (determined by *V*O_2max_ relative to BW and lean body mass (LBM)) within and between groups. In the PLA group, the *V*O_2max_ values before supplementation were 19.7 ± 3.5 mL/kg BW/min and 31.6 ± 5.2 mL/kg LBM/min, and after 12 weeks’ supplementation, they were 20.4 ± 3.7 mL/kg BW/min and 33.2 ± 5.5 mL/kg LBM/min. In the IG group, the *V*O_2max_ values before supplementation were 20.9 ± 4.1 mL/kg BW/min and 33.2 ± 6.9 mL/kg LBM/min, and after 12 weeks’ supplementation, they were 21.8 ± 3.5 mL/kg BW/min and 34.5 ± 6.8 mL/kg LBM/min (Figure 6).

## 4. Discussion

This is the first study that has investigated the effects of IG kernel extract supplementation on metabolism, adiposity indices, redox status, inflammation, adipocytokines, RTL, and aerobic capacity in overweight or obese individuals. We hypothesized that 12-week ingestion of IG kernel extract would improve these variables in these participants. Although our results did not support the entire hypothesis, we found evidence that supported some of it: we found beneficial effects in respect of antioxidant and hormonal responses to IG supplementation.

As hypothesized, we found increased concentrations of plasma vitamin C or ascorbic acid, a non-enzyme antioxidant, at 6 and 12 weeks after taking IG in overweight or obese participants (Figure 4b). The significant increase in plasma vitamin C concentration in the IG group is comparable with an in vitro study showing that IG extract exhibited antioxidant activity, which is a property of vitamin C. Although Mateus-Reguengo et al. (2020) found that IG fruit mesocarp contains various phytochemicals and ascorbic acid concentrations that are higher than some other vitamin-C-rich fruits [21], no study has reported the vitamin C content in IG kernel extract. This property, therefore, remains to be explored. Nonetheless, ours is the first study demonstrating that supplementation of IG kernel extract resulted in increased plasma vitamin C concentration (Figure 4b). Moreover, the improved antioxidant activity resulting from IG supplementation in this study was consistent with a previous IG study in human peripheral blood mononuclear cells [23]. The authors assessed the radical-quenching abilities of the IG extract compared with ascorbic acid at different concentrations using the 2,2-diphenyl-1-picrylhydrazyl (DPPH) method [23]. They found that IG extract exhibits almost 2 times the scavenging activity compared to ascorbic acid. However, no previous studies have investigated the antioxidant activity of IG in both animals and humans. Thus, this is the first study to demonstrate the antioxidant properties of IG extract in humans. Nonetheless, IG supplementation in this study had no beneficial effect on lipid peroxidation indicated by plasma MDA concentration (Figure 4a). MDA concentration may not be sensitive enough to measure the small changes caused by the IG supplement. F_2_-isoprostanes (F_2_-IsoPs) are reliable markers for assessing lipid peroxidation in vivo [36]. They can provide information by which to localize and quantify the specific oxidative stress. Despite this information, the utility of F_2_-IsoPs is highly limited because their use is costly, requiring gas/liquid chromatography coupled with mass spectroscopy techniques (HPLC/GC-MS). Thus, we did not measure F_2_-IsoPs because the method was not covered by the research grant. The other reason for the absent beneficial effect on lipid peroxidation is that IG extract may influence other oxidant mechanisms, e.g., oxidative DNA damage. Measuring a marker of oxidative DNA damage, i.e., 7-hydro-8-oxo-2′-deoxyguanosine [36], may disclose the effect of IG extract on this oxidant mechanism.

The other mechanism of IG supplementation that supports the hypothesis of this study is the hormonal response. The related hormones are serum leptin and adiponectin, which are adipokines secreted from adipocytes in many tissues, such as adipose tissue and skeletal muscle [37,38]. One of their important roles is energy regulation [38,39]. Adiponectin, the most abundant adipokine, plays a crucial role in obesity-related diseases, including metabolic syndrome [40]. We found that adiponectin concentrations were maintained in the IG group throughout the supplementation period. Except at 6 weeks in the IG group, the hormone concentration was slightly lower (8.7%) than the baseline. Importantly, we found greater adiponectin concentrations in the IG group compared to the PLA group after 3 weeks of supplementation, whereas those in the PLA group were dramatically lower than the baseline throughout the supplementation (Figure 3b). In contrast, leptin, which regulates food intake and body weight, and plays a role in proinflammatory immune responses and lipolysis [38], did not change as a result of IG supplementation in this study. However, leptin concentrations in the IG group were maintained, while those in the PLA group decreased throughout the supplementation. This may indicate a beneficial effect of IG supplementation that prevents a decrease in leptin levels (Figure 3a).

As aforementioned, the IG kernel extract obviously improved plasma vitamin C and maintained serum adiponectin concentrations in female overweight or obese individuals in this study. Unfortunately, we did not observe a significant effect of IG supplementation on other outcomes, including the fat oxidation rate (Table 4), adiposity indices (Table 3), circulating metabolic markers (Table 5), inflammation (Table 5), RTL (Figure 5), and aerobic capacity (Figure 6). This may be due to the insufficient duration or dose of the supplementation. Furthermore, apart from plasma vitamin C and adiponectin concentrations, there are more factors that influence these outcomes, such as daily diet and physical activity (Table 2). Therefore, the plasma vitamin C and adiponectin concentrations are not potent enough to cause changes in other outcomes. However, it may be worth discussing the correlation between plasma vitamin C and adiponectin and these outcomes.

Firstly, plasma vitamin C has been shown to be inversely related to central obesity indicated by waist-to-hip circumference ratio [41]. In addition, plasma vitamin C has been shown to be inversely related to body weight. This is supported by a study by Lee (2019), who found that ascorbic acid (1% *w/w* for 15 weeks) reduced visceral obesity [42]. Hence, plasma vitamin C seems to have an impact on the fat oxidation rate and body fat. Secondly, the plasma vitamin C concentration was negatively correlated with FBG in patients with type 2 diabetes mellitus [43]. Thirdly, although animal studies and observational trials suggest that vitamin C concentrations may correlate negatively with concentrations of TG, TC, and LDL-c, and positively correlate with HDL-c [44], inadequacies in trial design and significant variations may limit the conclusions. Fourthly, serum vitamin C concentrations have been shown to be negatively related to plasma concentrations of the inflammatory marker CRP [45]. Fifthly, plasma vitamin C concentrations are associated with longer leukocyte telomere length in healthy elderly people [46]. Lastly, lymphocyte ascorbic acid concentration was found to be positively associated with a high *V*O_2max_ or aerobic capacity [47]. This may be due to its antioxidant activity, which may protect against mitochondrial damage from the produced oxidants, leading to an effective oxidation process [48]. In addition to being a cofactor involved in carnitine biosynthesis, which produces a molecule (L-carnitine) required for the transport of fatty acids into mitochondria for the generation of metabolic energy [49], plasma vitamin C could have improved aerobic capacity in this study.

In addition, the correlation between serum adiponectin and other outcomes is interesting to explore. Previous studies in humans and rodents have shown that adiponectin has insulin-sensitizing, anti-inflammatory, and anti-atherogenic effects, and, in certain settings, also reduces body weight [37]. Interestingly, Broer et al. (2014) found that adiponectin showed a borderline significant association with RTL [50]. This appeared to be confirmed by a single study, and when adiponectin was removed, this association disappeared. Recently, Lendeckel et al. (2022) demonstrated the same direction of adiponectin and aerobic capacity regarding lifestyle modifications in obese patients with chronic kidney disease [51].

We have mentioned that the absence of the effects of IG supplementation on the fat oxidation rate, adiposity indices, circulating metabolic markers, inflammation, telomere length, and aerobic capacity may be due to the insufficient duration or dose of the supplementation. Therefore, further studies on IG supplementation over a longer period such as 6–12 months may show more definitive results. The dose could also be increased to 350 mg/day. However, this seems an unlikely option because there were side effects in participants in the IG group, including nausea (6 cases), intestinal constipation (1 case), and thirst (1 case). Although the side effects were not severe and resolved quickly, we should not put the participants at risk. Interestingly, we observed beneficial side-effects such as feeling full (2 cases) and better excretion (1 case). This suggests that a few participants may have experienced caloric intake reduction induced by the IG supplementation. Another method of determining the beneficial effect of IG supplementation on the other outcomes could be to add different extracts or other weight-reduction nutrients such as whey protein to the IG seed extract [52]. A previous study demonstrated that a combination of *Coleus forskohlii* root extract, citrus fruit extract, dihydrocapsiate, and red pepper fruit extract contributed to reducing body weight and fat gain under a high-fat diet [52]. Other studies in respect of traditional African medicine showed a weight-loss effect of African walnuts in Wistar rats [53] and an in vitro study of *Gnidia glauca* [54] and *Vernonia mespilifolia* Less. demonstrated anti-obesity and antioxidant effects [14].

The strength of our study lies in it including a body weight monitoring phase before the supplementation phase. This ensured that all participants had normal metabolism determined by maintaining their BW within a four-kilogram range. Furthermore, participants were tested for pregnancy to confirm a non-pregnant status. They also performed the experiment during the follicular phase of the menstrual cycle, during which there is a steady and low level of estrogen. This condition decreased the possibility of different concentrations of estrogen, which has a lipolytic effect [55,56]. Thus, we eliminated the confounding factor of estrogen influencing the fat oxidation rate of the participants. These factors confirm the high validity and reliability of our experiment. However, there were some limitations. Firstly, all the participants in the study were women. There is evidence that there may be gender differences in terms of the variables of this study [57,58]. Another limitation is the oxidative stress marker, i.e., MDA, which was discussed above. In addition, one might observe that compared with the PLA group, the IG group consumed less daily fat before and 3 weeks after the supplementation and consumed less protein throughout the supplementation. However, the participants had similar intakes of CHO, vitamin C, and energy, and similar energy expenditure, throughout the supplementation phase. The lack of differences in dietary vitamin C, energy intake, and energy expenditure between groups confirms that daily diet does not have any influence on the outcomes in this study. Further studies on IG supplementation over a longer period, such as 6–12 months, are warranted to confirm these findings and clarify any other beneficial effects. The reason for this speculation is based on an article on soy isoflavone supplementation [59], which has antioxidant activity similar to the vitamin C [60] found in our study. They reported that soy isoflavone supplementation could significantly decrease body weight (WMD, −0.506; 95%CI: −0.888 to −0.124; *p* = 0.009) over a shorter duration (<6 month) and could significantly decrease blood glucose (WMD, −0.270; 95%CI: −0.430 to −0.110; *p* = 0.001) over a longer duration (≥6 month) in postmenopausal women [59]. Regarding telomere length, Tsoukalas et al. (2019) reported that they took 6 to 12 months to find the beneficial effects of nutraceutical supplements on the telomere length in healthy subjects [61].

## 5. Conclusions

Our findings suggest that daily 300 mg IG kernel extract supplementation for 12 weeks improves plasma vitamin C and serum adiponectin concentrations without any changes on other variables. The benefit effects of IG might be an alternative solution to health and prevention of noncommunicable disease related to overweight/obesity.

## Figures and Tables

**Figure 1 nutrients-14-04646-f001:**
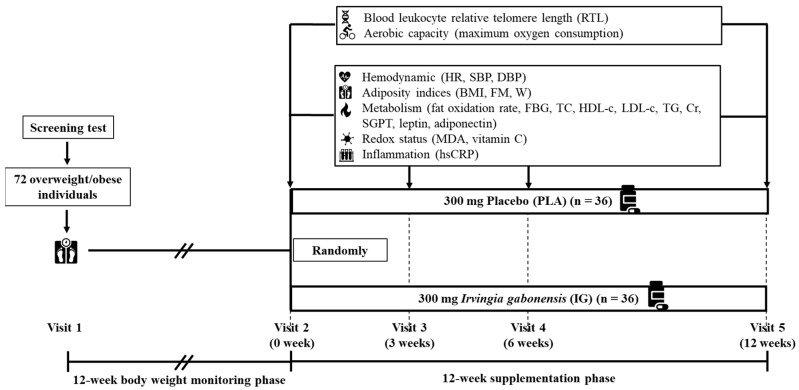
Protocol of this study. Abbreviations: HR, heart rate; SBP, systolic blood pressure; DBP, diastolic blood pressure; BMI, body mass index; FM, fat mass; W, waist circumference; FBG, fasting blood glucose; TC, total cholesterol; HDL-c, high-density lipoprotein cholesterol; LDL-c, low-density lipoprotein cholesterol; TG, triacylglycerol; Cr, creatinine; SGPT, serum glutamate pyruvate transaminase; MDA, malondialdehyde; hsCRP, high-sensitivity C-reactive protein.

**Figure 2 nutrients-14-04646-f002:**
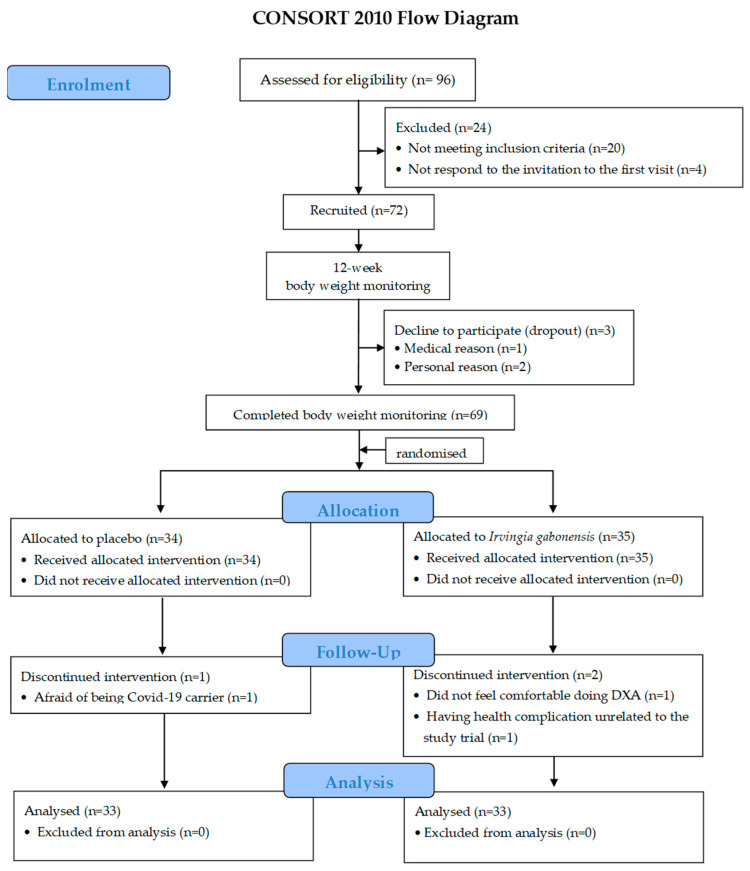
CONSORT flow diagram.

**Figure 3 nutrients-14-04646-f003:**
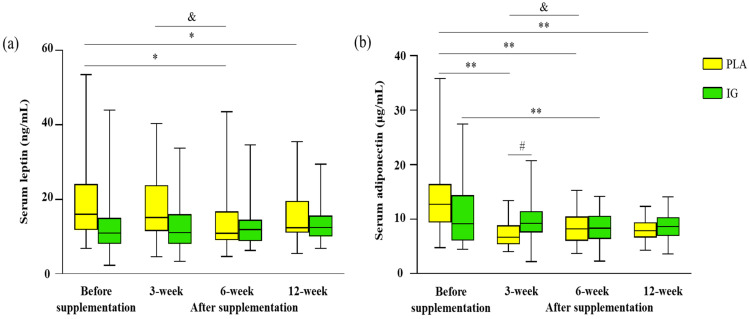
Serum leptin concentration (ng/mL) (**a**) and serum adiponectin concentration (µg/mL) (**b**) before and after supplementation in both groups (n = 33 each group). ^#^
*p* < 0.05 presents significant difference between groups at the same time point; * *p* < 0.05, ** *p* < 0.01, present significantly different from before supplementation within group. ^&^
*p* < 0.05 presents significantly different from 3 weeks after supplementation within group. The boxplots of the non-normal distributions present median and interquartile ranges. Abbreviation: PLA, placebo; IG, *Irvingia gabonensis*.

**Figure 4 nutrients-14-04646-f004:**
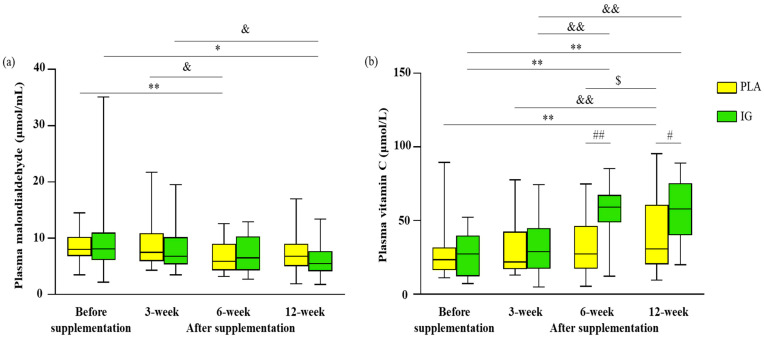
Redox status presented by plasma malondialdehyde (MDA) concentration (µmol/mL) (**a**) and plasma vitamin C concentration (µmol/L) (**b**) before and after supplementation in both groups (n = 33 each group). ^#^
*p* < 0.05, ^##^
*p* < 0.01 present significant differences between groups at the same time; * *p* < 0.05, ** *p* < 0.01 present significant differences from before supplementation within group (*p* < 0.05). ^&^
*p* < 0.05, ^&&^
*p* < 0.01 present significant differences for after 3 weeks of supplementation within group. ^$^
*p* < 0.05 presents significant difference for after 6 weeks of supplementation within group. The boxplots of the non-normal distributions are presented as median and interquartile ranges. Abbreviations: PLA, placebo; IG, *Irvingia gabonensis*.

**Figure 5 nutrients-14-04646-f005:**
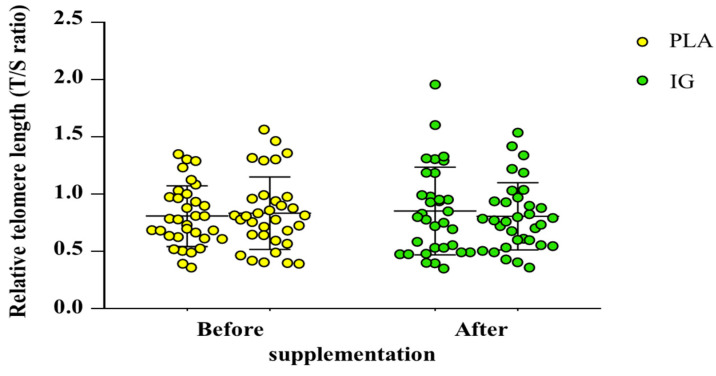
Blood leukocyte relative telomere length (RTL) of the participants before and after supplementation phase (n = 33 each group). Abbreviations: PLA, placebo; IG, *Irvingia gabonensis*.

**Figure 6 nutrients-14-04646-f006:**
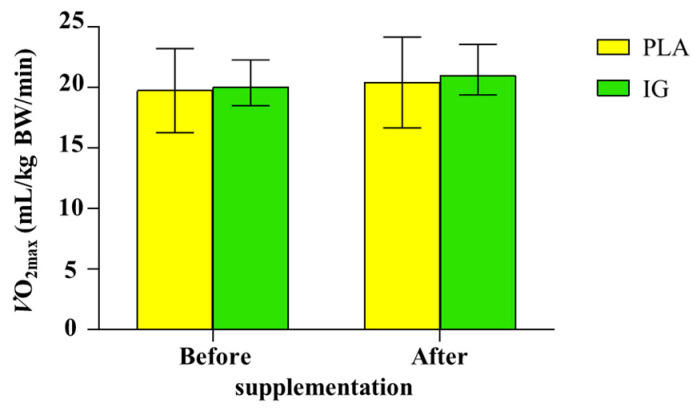
Aerobic capacity (mL/kg BW/min) of the participants tested using incremental exercise test. The bar graph presents mean ± SD (n = 33 each group). Abbreviations: PLA, placebo; IG, *Irvingia gabonensis*.

**Table 1 nutrients-14-04646-t001:** Characteristics of participants (who completed supplementation phase; n = 66) before and after BW monitoring phase.

	Body Weight Monitoring Phase	
	Before (Visit 1) (n = 66)	After (Visit 2) (n = 66)
Age (years)	44.7 ± 6.7	44.7 ± 6.7
Gender (women/men)	66/0	66/0
BW (kg)	61.1 ± 5.4	61.2 ± 5.5
Ht (m)	1. 6 ± 0.1	1. 6 ± 0.1
BMI (kg/m^2^)	24.7 (24.1–26.4)	25.1 (24.1–26.5)
W (cm)	86.4 ± 6.1	88.0 ± 6.3
H (cm)	99.2 ± 4.9	99.8 ± 4.6
W/H	0.86 (0.84–0.91)	0.88 (0.84–0.92)
RHR (/min)	68.1 ± 8.5	65.5 ± 7.9
SBP (mmHg)	119.3 ± 12.9	119.6 ± 14.2
DBP (mmHg)	75.0 ± 9.8	75.9 ± 9.1
MAP (mmHg)	85.5 (78.5–91.5)	85.0 (79.0–90.0)
FBG (mg/dL)	92.9 (84.0–104.0)	87.0 (84.0–93.0)
TC (mg/dL)	186.7 ± 33.5	189.7 ± 37.1
HDL-c (mg/dL)	50.4 ± 14.2	52.7 ± 11.6
LDL-c (mg/dL)	110.4 ± 32.3	114.0 ± 34.0
HDL-c/LDL-c ratio	0.43 (0.33–0.60)	0.47 (0.39–0.57)
TC/HDL-c ratio	3.78 (3.15–4.55)	3.50 (3.06–3.97)
TG (mg/dL)	105.5 (85.1–137.0)	100.9 (71.0–140.0)
Cr (mg/dL)	0.7 ± 0.1	0. 8 ± 0.1
SGPT (IU/L)	14.7 (11.7–18.1)	14.0 (11.6–18.1)

All data are represented as mean ± SD for normal distribution data or median (interquartile range, IQR) for non-normal distribution data (n = 66). Abbreviations: BW, body weight; Ht, height; BMI, body mass index; W, waist circumference; H, hip circumference; W/H, waist and hip circumference ratio; RHR, resting heart rate; SBP, systolic blood pressure; DBP, diastolic blood pressure; MAP, mean arterial pressure; FBG, fasting blood glucose; TC, total cholesterol; HDL-c, high-density lipoprotein cholesterol; LDL-c, low-density lipoprotein cholesterol; TG, triacylglycerol; Cr, creatinine; SGPT, serum glutamate pyruvate transaminase.

**Table 2 nutrients-14-04646-t002:** Energy intake and expenditure before and after supplementation for 3, 6, and 12 weeks in both groups.

	PLA (n = 33)	IG (n = 33)
CHO intake (g)		
Before	257.5 ± 61.8	253.5 ± 63.8
After 3 weeks	249.4 ± 78.19	264.5 ± 61.7
After 6 weeks	243.1 ± 68.2	253.7 ± 58.4
After 12 weeks	267.8 ± 72.0	239.2 ± 59.0
Fat intake (g)		
Before	49.8 ± 21.3	37.9 ± 14.5 #
After 3 weeks	51.9 ± 22.7	36.2 ± 11.5 ##
After 6 weeks	44.8 ± 15.6	36.8 ± 14.0
After 12 weeks	48.5 ± 17.1	37.4 ± 11.1
Protein intake (g)		
Before	79.5 ± 19.3	66.8 ± 15.4 #
After 3 weeks	82.4 ± 27.9	63.5 ± 13.3 ##
After 6 weeks	79.3 ± 20.6	65.4 ± 13.3 ##
After 12 weeks	82.3 ± 22.9	61.3 ± 12.9 ##
Vitamin C intake (mg/day)		
Before	49.6 (32.5–95.6)	48.5 (34.8–74.9)
After 3 weeks	71.6 (32.2–87.7)	43.0 (26.4–59.3)
After 6 weeks	66.2 (33.5–84.8)	48.4 (39.0–90.1)
After 12 weeks	49.8 (26.8–68.7)	48.5 (26.9–75.0)
Energy intake (kcal/day)		
Before	1796.4 ±440.7	1622.2 ± 329.5
After 3 weeks	1794.2 ± 488.4	1637.8 ± 299.4
After 6 weeks	1692.9 ± 381.5	1607.5 ± 296.2
After 12 weeks	2122.5 ± 1734.0	1538.6 ± 294.7
Energy expenditure (kcal/day)		
Before	1376.8 ± 260.3	1461.2 ± 218.1
After 3 weeks	1444.7 ± 296.6	1462.0 ± 274.1
After 6 weeks	1404.3 ± 370.1	1470.0 ± 278.9
After 12 weeks	1435.5 ± 332.3	1487.9 ± 284.0

All data are represented as mean ± SD for normal distribution data or median (interquartile range, IQR) for non-normal distribution data (n = 66). All data were analyzed using the general estimated equation (GEE) (*p* < 0.05), and a significant difference was specified via a post hoc analysis using Bonferroni correction. ^#^
*p* < 0.05, ^##^
*p* < 0.01 present significant differences between groups at the same time point. Abbreviation: PLA, placebo; IG, *Irvingia gabonensis*.

**Table 3 nutrients-14-04646-t003:** Body composition before and after supplementation for 3, 6, and 12 weeks in both groups.

	PLA (n = 33)	IG (n = 33)
BW (kg)		
Before	59.7 (56.8–64.8)	61.1 (57.3–63.6)
After 3 weeks	60.5 (56.9–65.1)	62.0 (57.3–64.1)
After 6 weeks	60.2 (56.9–65.3)	62.0 (57.5–63.5)
After 12 weeks	60.0 (56.6–65.6)	62.4 (57.3–64.0)
BMI (kg/m^2^)		
Before	25.3 (24.1–27.0)	24.7 (24.1–25.6)
After 3 weeks	25.8 (23.9–27.0)	24.8 (24.1–26.0)
After 6 weeks	25.7 (24.0–26.2)	24.9 (24.2–26.0)
After 12 weeks	25.3 (24.0–27.3)	24.7 (24.2–26.0)
W (cm)		
Before	88.6 ± 6.8	87.3 ± 5.8
After 3 weeks	88.4 ± 7.0	86.9 ± 5.7
After 6 weeks	88.2 ± 6.9	86.3 ± 5.4
After 12 weeks	87.6 ± 7.0	86.6 ± 6.1
FM (%)		
Before	36.4 ± 2.8	35.5 ± 3.1
After 3 weeks	36.7 ± 2.7	35.8 ± 3.1
After 6 weeks	36.7 ± 2.9	35.9 ± 3.1
After 12 weeks	36.8 ± 2.8	35.9 ± 3.1
FM (kg)		
Before	23.1 ± 3.6	21.9 ± 3.9
After 3 weeks	23.3 ± 3.5	22.2 ± 2.9
After 6 weeks	23.3 ± 3.6	22.2 ± 2.8
After 12 weeks	23.2 ± 3.6	22.1 ± 2.6

All data are represented as mean ± SD or median (interquartile range, IQR) based on normal distribution (n = 66). All data were analyzed using a general estimated equation (GEE) (*p* value < 0.05), and a significant difference was specified via a post hoc analysis using Bonferroni correction. Abbreviations: PLA, placebo; IG, *Irvingia gabonensis*; BW, body weight; BMI, body mass index; W, waist circumference; FM, fat mass.

**Table 4 nutrients-14-04646-t004:** Fat oxidation rate before and after supplementation for 3, 6, and 12 weeks in both groups.

	PLA (n = 33)	IG (n = 33)	Mean Difference (95% CI)
Fat oxidation rate (mg/min)			
Before	45.7 ± 22.2	43.6 ± 20.2	2.08 mg/min; 95%CI, −8.34, 12.51
After 3 weeks	43.0 ± 22.8	43.6 ± 25.0	−0.58 mg/min; 95%CI, −12.34, 11.18
After 6 weeks	44.5 ± 24.0	37.7 ± 15.3	6.83 mg/min; 95%CI, −3.07, 16.72
After 12 weeks	48.3 ± 19.9	40.5 ± 25.8	7.81 mg/min; 95%CI, −3.52, 19.14

All data. All data were analyzed using a general estimated equation (GEE) (*p* < 0.05), and a significant difference was specified via a post hoc analysis using Bonferroni correction. All data are represented as mean ± SD based on normal distribution (n = 66). Abbreviation: PLA, placebo; IG, *Irvingia gabonensis*.

**Table 5 nutrients-14-04646-t005:** FBG and plasma lipid profile before and after supplementation for 3, 6, and 12 weeks in both groups.

	PLA (n = 33)	IG (n = 33)
FBG (mg/dL)		
Before	87.0 (82.0–92.0)	87.0 (85.0–93.0)
After 3 weeks	85.0 (82.0–92.0)	86.0 (82.0–94.0)
After 6 weeks	86.0 (83.0–90.0)	89.0 (82.0–94.0)
After 12 weeks	88.0 (82.0–91.0)	89.0 (86.0–91.0)
TC (mg/dL)		
Before	182.0 (166.0–209.0)	191.0 (166.0–191.0)
After 3 weeks	186.0 (170.0–217.0)	187.0 (170.0–212.0)
After 6 weeks	192.0 (176.0–206.0)	190.0 (167.0–207.0)
After 12 weeks	177.5 (166.0–207.0)	192.0 (169.0–220.0)
HDL-c (mg/dL)		
Before	51.7 (42.6–58.4)	54.2 (48.2–62.8)
After 3 weeks	47.1 (40.5–54.6)	52.8 (44.5–57.5)
After 6 weeks	47.7 (41.9–55.5)	52.5 (44.7–60.9)
After 12 weeks	53.6 (43.1–58.6)	47.0 (41.0–57.2)
LDL-c (mg/dL)		
Before	111.6 (91.9–133.6)	108.6 (98.9–139.0)
After 3 weeks	112.1 (101.7–140.2)	106.4 (94.7–137.4)
After 6 weeks	116.7 (105.0–142.9)	115.8 (93.7–132.7)
After 12 weeks	112.1 (83.5–137.5)	112.3 (94.4–142.4)
HDL-c/LDL-c ratio		
Before	0.5 (0.4–0.6)	0.6 (0.4–0.6)
After 3 weeks	0.5 (0.3–0.5)	0.5 (0.3–0.6)
After 6 weeks	0.4 (0.3–0.5)	0.4 (0.4–0.6)
After 12 weeks	0.5 (0.3–0.6)	0.4 (0.3–0.6)
TC/HDL-c ratio		
Before	3.5 (3.1–4.0)	3.5 (2.8–3.9)
After 3 weeks	3.6 (3.3–5.3)	3.7 (3.1–4.4)
After 6 weeks	3.9 (3.4–4.5)	3.8 (3.2–4.2)
After 12 weeks	3.4 (3.1–4.4)	3.8 (3.3–4.8)
TG (mg/dL)		
Before	97.7 (82.5–140.0)	106.0 (69.0–134.0)
After 3 weeks	95.0 (69.0–141.0)	112.0 (88.9–153.0)
After 6 weeks	111.0 (84.2–145.0)	105.0 (81.0–170.0)
After 12 weeks	89.2 (73.4–126.0)	119.0 (91.9–178.0) *
hsCRP (mg/dL)		
Before	1.6 (0.8–2.6)	1.3 (0.8–1.8)
After 3 weeks	1.3 (0.8–2.2)	1.2 (0.8–1.9)
After 6 weeks	1.9 (1.1–2.8)	1.2 (0.9–2.2)
After 12 weeks	1.5 (0.8–2.2)	1.2 (0.7–1.9)

All data were analyzed using a general estimated equation (GEE) (*p* < 0.05), and a significant difference was specified via a post hoc analysis using Bonferroni correction. * *p* < 0.05 presents a significant difference from before the supplementation phase within the group. Abbreviation: PLA, placebo; IG, *Irvingia gabonensis*; FBG, fasting blood glucose; TC, total cholesterol; HDL-c, high-density lipoprotein cholesterol; LDL-c, low-density lipoprotein cholesterol; TG, triacylglycerol. All data are represented as median (interquartile range, IQR) based on normal distribution (n = 66).

## Data Availability

Data can be provided upon request, and the full trial protocol can be accessed at http://www.thaiclinicaltrials.org/show/TCTR20200803005 (accessed on 2 August 2020).

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
