# Peer review of "Effects of Irvingia gabonensis Extract on Metabolism, Antioxidants, Adipocytokines, Telomere Length, and Aerobic Capacity in Overweight/Obese Individuals"

_nutrients, 2022, doi:10.3390/nu14214646_

Round 1

Reviewer 1 Report

Rujira et al., explored the effect of Irvingia gabonensis extract on different metabolic and genetic risk factors in overweight and obese individuals. The manuscript might be interesting. Following modifications are suggested:

1. Title is misleading and not in line with the abstract i.e., the information provided in title is not covered in abstract (e.g., telomere length etc.)

2. Introduction needs revision. Why authors selected rvingia gabonensis extract not any other plant extract? Stating that no one studied before is not a satisfactory statement. The rationale should be supported by some scientific facts.

3. It is suggested that authors discuss some related plant extracts for a better comparison.

4. subsection 2.1: mention number of participants.

5. "7 mL whole blood was drawn from antecubital vein" why 7 mL not less than 7 mL? Isn't it a very sample volume?

8. "Though we also found beneficial symptoms such as feeling full (2 cases)" Please rewrite this sentence for better understanding. 

9. Please cite recent literature.

10. What was the criterion to decide placebo?

Reviewer 2 Report

The work entitled “Effects of Irvingia gabonensis extract on metabolism, antioxi-dants, adipocytokines, telomere length, and aerobic capacity in overweight/obese participants” attempts to evaluate the health benefits of Irvingia gabonensis on overweight/obese participants. The study is interesting, and the results are clear, but I have some concerns:

1. Has the kernel extract of Irvingia gabonensis been evaluated for safety before the study? If you have such date to support this, please provided.

2. In the introduction, you talked about telomeres in detail. But for your results, the telomeres length did not differ between groups (PLA and IG) or before and after treatment. So, I suggested to just give a brave introduction about telomeres. Instead, the roles of Vitamin C and hormones (leptin and adiponectin) should be introduced in detail.

3. Lines88-90: What does “improves these outcomes” mean? Please rewrite to make the meaning of the expression clearer.

4. Table 1: What do the values in parentheses mean? Please give an explanation.

5. Table 2: How did you get this date? Please describe it in the materials and methods. What do the values in parentheses mean? Please give an explanation. / Whether there were significant differences in various test indexes between different groups?

6. Talbe3: What do the values in parentheses mean? Please give an explanation. Whether there were significant differences in various test indexes between different groups?

7. Table 4: How did you calculate the rate of fat oxidation? Please describe it in the materials and methods.

8. Figure 3/Figure 4: What statistical methods were used to process the data? I don't think these differences are statistically reliable.

9. Figure 5: Relative telomere length was found not significantly different within and between both groups. Could this result indicate that IG did not have any effect on telomere length? If so, what is the significance of this result? Could this support the conclusion of your article?

10. 3.5 Aerobic capacity: Please give a conclusion of this part.

11. Lines 461-467: The reference [33] was just test the antioxidant activity of IG. I don’t think it’s enough to support you conclusion “The increased plasma vitamin C concentration is likely to be due to IG supplementation”

12. In the discussion, you discussed the vitamin C and adiponectin again and again? How about the other results? I suggested to added discussion about the other results.

Round 2

Reviewer 1 Report

Revised version is much improved.

Author Response

Dear reviewer 1

We thank the reviewer 1 very much for the compliment that our revised version is much improved and not having more comment.

Moreover, we appreciate for the quick review.

Best regards

Naruemon Leelayuwat
